# NEURAL PHYLOGENY: FINE-TUNING RELATIONSHIP DETECTION AMONG NEURAL NETWORKS

**Runpeng Yu, Xinchao Wang**[*]

National University of Singapore
`r.yu@u.nus.edu, xinchao@nus.edu.sg`

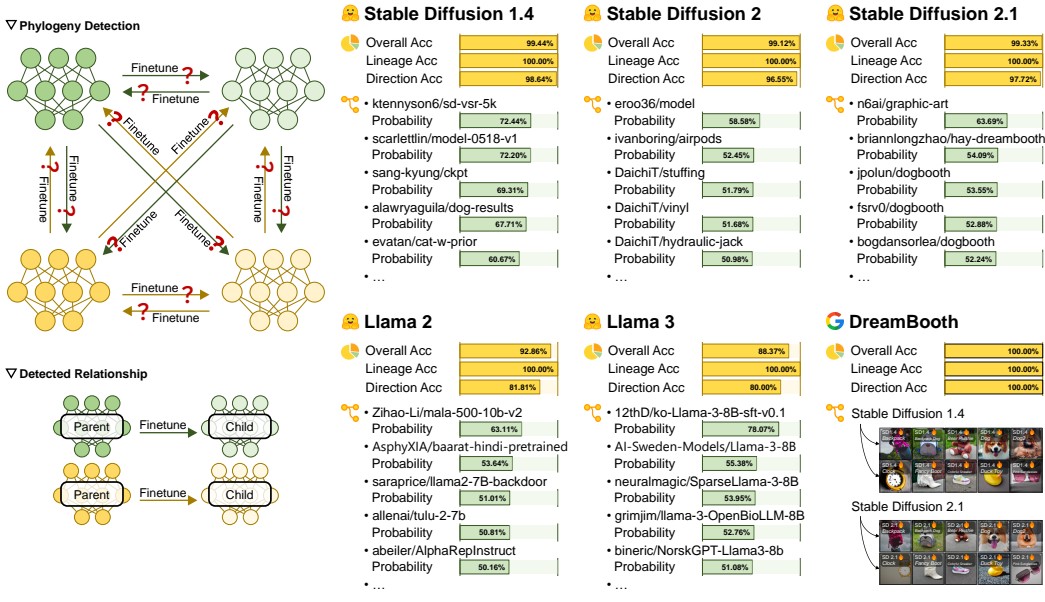

(a) Task illustration

(b) Neural phylogeny detection for Stable Diffusion and Llama.

Figure 1: Neural phylogeny detection showcase. Fig. 1a shows the idea of neural phylogeny detection. Fig. 1b presents the detection performance among Stable Diffusion models and among Llama models. The detailed experimental setup and discussion are included in the Sec. 4 and Appendix A.

## ABSTRACT

Given a collection of neural networks, can we determine which are parent models and which are child models fine-tuned from the parents? In this work, we strive to answer this question via introducing a new task termed as *neural phylogeny* detection, aimed at identifying the existence and direction of the fine-tuning relationship. Specifically, neural phylogeny detection attempts to identify all parent-child model pairs and determine, within each pair, which model is the parent and which is the child. We present two approaches for neural phylogeny detection: a learning-free method and a learning-based method. First, we propose to use the norm of the network parameters to infer fine-tuning directions. By integrating this metric with traditional clustering algorithms, we propose a series of efficient, learning-free neural phylogeny detection methods. Second, we introduce a transformer-based neural phylogeny detector, which significantly enhances detection accuracy through a learning-based manner. Extensive experiments, ranging from shallow fully-connected networks to open-sourced Stable Diffusion and LLaMA models, progressively validate the effectiveness of both methods. The results demonstrate the reliability of both the learning-free and the learning-based approaches across various learning tasks and network architectures, as well as their

---

[*]Corresponding Author.

ability to detect cross-generational phylogeny between ancestor models and their fine-tuned descendants.

# 1 INTRODUCTION

Recently, the field of deep learning has experienced transformative evolutions. First, traditional "training from scratch" paradigm has shifted to the "pre-training-fine-tuning" approach, which allows for more efficient model development, meanwhile, creating a substantial dependency on models and their derivatives. Additionally, with the expansion of the deep learning research community and commercial success, numerous open-source model repositories have been established, and models produced by "machine learning as a service" are accumulating. These advancements signify the emergence of a complex, interconnected network of deep learning models. Studying this network can help us better understand the inheritance of knowledge between models and the propagation of bias and unfairness, also facilitating the management of intellectual property. However, such research relies on the ability to first unveil this underlying network of neural networks.

In this work, we define and investigate the task of neural phylogeny detection, as illustrated in Fig. 1a. Specifically, given a parent neural network $f_p$, the model $f_c$, which is fully fine-tuned from it, is referred to as its child model. Considering a set of neural networks, which includes several parent models and their fine-tuned child models, the goal of neural phylogeny detection is to identify all parent-child model pairs within this set and determine which model is the parent model and which is the child model in each pair.

We propose two approaches to address the neural phylogeny detection task: a learning-free method and a learning-based method.

To develop the learning-free method, we first investigate how to determine the direction of fine-tuning. Specifically, given two models with a fine-tuning relationship, we aim to identify which is the parent model and which is the child model. Through theoretical analysis, we discover that the norm of neural network parameters serves as a simple yet effective method for determining the direction of fine-tuning. We validate this theoretical result on both simple ResNet architecture, and more complex models, including Stable Diffusion and LLaMA models. For instance, in the case of Diffusion Models, using parameter norms enables us to identify the fine-tuning direction with an accuracy exceeding 95%. Building on this insight, we propose a learning-free method for phylogeny detection.

In the learning-free method, phylogeny detection is treated as a clustering task. Parent models and their fine-tuned child models form clusters, with the parent identified as the centroid. Using clustering to achieve phylogeny detection involves two sub-tasks: clustering models into clusters and identifying the parent model within each cluster. The clustering sub-task can be accomplished using traditional clustering algorithms, while the task of identifying the parent model is handled by the norm of the parameters. We explore two workflows to achieve these two sub-tasks. The first workflow performs clustering first, and the parent in each cluster is identified after the clustering algorithm is complete. The second approach integrates the parent model identification into the clustering algorithm, and by the end of clustering, the parent models within each cluster are also identified. We apply these two strategies to commonly used clustering algorithms, resulting in a series of learning-free methods for efficient neural phylogeny detection.

In the learning-based approach, we frame neural phylogeny detection as a directed graph learning task. Each model is treated as a node, with fine-tuning relationships forming directed edges between parent and child models. We design a transformer-based detector to predict the adjacency matrix of this graph. Unlike the learning-free method, which relies on a specific metric, the learning-based approach learns to infer the existence and the direction of finetuning, achieving better performance.

The contributions of this work are:

- We investigate the novel task of neural phylogeny detection, aiming to identify parent-child model pairs among neural networks and determine the fine-tuning direction.
- We propose two approaches: a learning-free method that is convenient to implement and a learning-based method that offers higher prediction accuracy.
- We demonstrate the effectiveness of the proposed methods on various deep model architectures from shallow fully-connected nerual network to large open-source models; under different learning

settings, including few-shot learning and imbalanced data learning; and for cross-generational neural phylogeny detection.

## 2 RELATED WORKS

### 2.1 NEURAL LINEAGE AND DEEP MODEL IP PROTECTION

This work is most closely related to neural lineage detection in (Yu & Wang, 2024), which identifies from which parent model a given child model has been fine-tuned. Neural phylogeny detection is a more challenging task and has substantial differences compared with neural lineage detection. (1) In neural lineage detection, the sets of parent models and the set of child models are distinct with no overlap, thus the direction of fine-tuning is known; whereas in neural phylogeny detection, all models are mixed together, and the direction of fine-tuning is to-be-determined. (2) In neural lineage detection, the finetuning dataset is accessible and the forward inference and gradient propagation of the models are required. Conversely, neural phylogeny detection is data-free and does not require the forward and backward of the model. (3) Neural lineage detection is a classification task, while in this work, neural phylogeny detection is treated as a clustering or graph structure learning task.

On the other hand, by incorporating information related to the training data, neural phylogeny detection can be simplified into neural lineage detection. For instance, knowledge of the input and labels from the dataset used for pretraining or fine-tuning would facilitate the differentiation between parent and child models through measures such as accuracy, loss values, or gradient magnitudes, effectively reducing the phylogeny detection to lineage detection.

Another related task is deep model IP protection, which is commonly addressed through model watermarking and model fingerprinting. Model watermarking identifies a stolen model via an embedded watermark (Uchida et al., 2017; Chen et al., 2019; Li et al., 2020b; 2022; Lou et al., 2021; Chen et al., 2021b; Lounici et al., 2021; Lao et al., 2022; Nagai et al., 2018; Zhao et al., 2021; Xie et al., 2021; Darvish Rouhani et al., 2019; Adi et al., 2018; Ye et al., 2023; Jing et al., 2021; 2023), whereas model fingerprinting discovers the source model and a stolen model based on their behaviors across a set of conferrable samples (Cao et al., 2021; Lukas et al., 2021; Peng et al., 2022; Li et al., 2021; Pan et al., 2021; Zhao et al., 2020; Yang et al., 2022; Wang & Chang, 2021). Deep IP protection also claims that IP protection methods are capable of identifying source model and its child models against the fine-tuning attack. However, in deep IP protection, fine-tuning is often conducted using minimal learning rates and few epochs to maintain the child model's performance on the original task (Darvish Rouhani et al., 2019; Jia et al., 2021; Li et al., 2020a; Cao et al., 2021; Pan et al., 2022; Fan et al., 2019; Lee et al., 2023; Adi et al., 2018), which differs significantly from the fine-tuning process in practice. By contrast, the fine-tuning in neural phylogeny is more aligned with practice, in which finetuning is method for transfer learning.

### 2.2 NEURAL NETWORK LINEARIZATION

In this work, we build our theoretical analysis upon the framework and conclusions of neural linearization. Unlike previous studies that applied neural linearization to pruning at initialization (Wang et al., 2020; Gebhart et al., 2021; Fang et al., 2023; Ma et al., 2023), training-free neural architecture search (Chen et al., 2021a; Wang et al., 2022; Mok et al., 2022), and training time prediction (Zancato et al.), our approach represents a new use of linearization theory, specifically for detecting fine-tuning direction. The theoretical foundation of neural linearization suggests that, as network width grows, the network can be well represented by its first-order Taylor expansion at initialization (Lee et al., 2019a). Further research indicates that these linear approximations can achieve comparable or even superior performance to their nonlinear counterparts (Ortiz-Jiménez et al., 2021; Arora et al., 2020). By leveraging neural linearization in our method, we explored which metrics exhibit monotonic changes during the fine-tuning process, allowing them to be used to determine the direction of fine-tuning.

### 2.3 GRAPH STRUCTURE LEARNING

In learning-based methods, we treat the problem of neural phylogeny detection as a novel application of graph structure learning. Unlike previous works that have applied graph structure learning to tasks like automatic diagnosis Cosmo et al. (2020) and protein structure prediction Jumper et al.

(2021), our approach specifically targets the identification of phylogenetic relationships in neural networks, which presents unique challenges and opportunities. Graph structure learning generally aims to identify valuable graph structures directly from data, resulting in undirected Chen & Wu (2022), weighted Chen & Wu (2022), or directed graphs (Yu et al., 2019). Prior research has explored aspects such as the interplay between structure learning and downstream tasks, the parameterization of adjacency matrices, training loss design, and post-processing methods like discretization and sparsification Liu et al. (2022); Chen et al. (2020); Fatemi et al. (2021); Franceschi et al. (2019); Jin et al. (2020); Wang et al. (2021); Yu et al. (2020). Our work differentiates itself by focusing on the adaptation of these techniques to capture evolutionary relationships between neural networks.

## 3 METHOD

### 3.1 NOTATION AND TASK

**Notation**. We use $f$ to denote a neural network and $\mathcal{D} = (\mathcal{X}, \mathcal{Y})$ to denote the training dataset, where $\mathcal{X} = \{\boldsymbol{x} : (\boldsymbol{x}, \boldsymbol{y}) \in \mathcal{D}\}$ and $\mathcal{Y} = \{\boldsymbol{y} : (\boldsymbol{x}, \boldsymbol{y}) \in \mathcal{D}\}$ are the inputs and the targets, respectively. We use $\boldsymbol{\theta}$ to denote the parameters of the neural network and present it in the vectorization form to facilitate notation. For example, we write $\boldsymbol{\theta} \in \mathbb{R}^{d_{\boldsymbol{\theta}} \times 1}$, where $d_{\boldsymbol{\theta}}$ is the number of the parameters in $f$. Given a set of neural network models, we use the superscripts to index the models. Specifically, $f^{(i)}$ is used to denote the $i$-th model in the set. The parameter of $f^{(i)}$ is denoted by $\boldsymbol{\theta}^{(i)}$. Given a matrix , let $\boldsymbol{z}_{i,:}$ and $\boldsymbol{z}_{i,j}$ to denote its $i$-th row and its element in the $i$-th row and $j$-th column, respectively. Given a directed graph $G$, let $e_{i,j}$ to denote the edge from the $i$-th node to the $j$-th node. Let $\mathbf{1}(\cdot)$ denote the indicator function.

**Task**. The objective of neural phylogeny detection is to first identify all parent-child pairs within a set of neural network models. Then, it seeks to determine the direction of fine-tuning between each detected parent and child pair. Specifically, given a set of neural networks, $\{f^{(i)}\}_{i=1}^{M}$, containing parent and their child models, neural phylogeny aims to find a binary matrix $\boldsymbol{A}$, where $\boldsymbol{A}_{ij} = 1$ if the $i$-th model $f^{(i)}$ is the parent of the $j$-th model $f^{(j)}$, and $\boldsymbol{A}_{ij} = 0$ otherwise. In this work, neural phylogeny detection is conducted in a completely data-free manner, meaning that we only access the parameters of the neural networks without any information related to their training data.

### 3.2 DIRECTION OF THE GRADIENT DESCENT

In this subsection, we investigate how a simple metric — the norm of neural network parameter — can be used to detect the direction of finetuning. We begin by evaluating the feasibility of this metric in a simplified theoretical setting, and conclude by validating that this simple metric can still achieve reasonable accuracy in predicting the direction of finetuning on open-source models such as Llama and Stable Diffusion.

Neural phylogeny detection comprises two sub-tasks: identifying parent-child pairs and determining the direction of fine-tuning, or, in other words, the direction of gradient descent. The detection of parent-child pairs can be effectively achieved using the similarity between models (Yu & Wang, 2024). Therefore, the crux of phylogeny detection is the determination of the direction of gradient descent.

We first demonstrate through the Prop. 1 below that, in theory, the norm of neural network parameters increases during the training process. Therefore, the norm of the parameter serves as a theoretically feasible metric for determining the finetuning direction. In Prop. 1, we follow the assumption and formulation in neural network linearization Lee et al. (2019b) to describe the dynamic of the neural network training. Previous works have shown that, for wide network, the linearized network can be used to effectively and efficiently approximate the dynamic of the original non-linear network in the gradient descent.

**Proposition 1.** *Let $f_t$ denote a neural network trained using gradient flow on the dataset $\mathcal{D} = (\mathcal{X}, \mathcal{Y})$ at time $t$. Let $\boldsymbol{\theta}_t \in \mathcal{R}^{d_{\boldsymbol{\theta}} \times 1}$ denote the vectorized parameters of $f_t$, where $d_{\boldsymbol{\theta}}$ is the number of parameters. Let $f_0$ denote the neural network at the initialization with initial parameters $\boldsymbol{\theta}_0$. Let $\eta$ denote the learning rate. Let $\Theta$ denote the neural tangent kernel at the initialization. Let $\epsilon$ denote the small term, whose specific definition is included in Appendix D.1. Then, with MSE loss, the squared*

Table 1: By comparing norm of the parameter, the parent model in a parent-child pair can be detected with high accuracy.

|  | Stable Diffusion 1.4 | Stable Diffusion 2 | Stable Diffusion 2.1 | Llama 2 | Llama 3 |
|---|---|---|---|---|---|
| Accuracy | 96.49 | 95.12 | 97.33 | 88.57 | 58.01 |

$\ell_2$ *distance between* $\boldsymbol{\theta}_t$ *and* $\boldsymbol{z}$ *can be approximated as,*

$$dist(\boldsymbol{\theta}_t, \boldsymbol{z}) \triangleq \frac{1}{d_{\boldsymbol{\theta}}}||\boldsymbol{\theta}_t - \boldsymbol{z}||_2^2 = \frac{1}{d_{\boldsymbol{\theta}}}||\nabla_{\boldsymbol{\theta}_0}f_0(\mathcal{X})^T\Theta^{-1}(I - e^{-\eta\Theta t})(f_0(\mathcal{X}) - \mathcal{Y})]||_2^2 + \epsilon, \quad (1)$$

*which increases over time.*

Prop. 1 demonstrates that the norm of parameters $\boldsymbol{\theta}_t$ is a function that increases over time $t$, which validates the feasibility of using such norm to determine the direction of neural network training. Furthermore, in the supplementary materials, we extend the above Prop. 1 to scenarios involving weight decay regularization. It can be proven that the norm of the parameters still increases during training, provided the weight decay coefficient is not large. This is consistent with practical scenarios, as a large weight decay coefficient can impair the performance of the network and is therefore not utilized.

To assess the practical usefulness of the theoretical results, we conducted two analyses. (1) We trained eight ResNet18 networks using different combinations of hyperparameters and fine-tuned them. Fig. 2 records the norm of their parameters during both the training and fine-tuning processes. The distance increases both during the pre-training phase and the subsequent fine-tuning phase, which validates the conclusion in Prop. 1. (2) We downloaded 3 parent Stable Diffusion models and 2 parent LLaMA models, along with their finetuned child models from HuggingFace. In total, we downloaded 366 Stable Diffusion models and 95 Llama models. We evaluated whether the norm of the parameter could effectively detect the fine-tuning direction among open-source models. Tab. 1 reports the performance, showing that the norm can consistently identify the finetuning direction with high accuracy. To mitigate the curse of dimensionality and reduce the computational costs, in this and subsequent experiments, we use a subset of the parameters selected based on a validation set composed of 10% of the models.

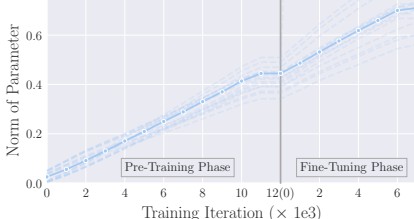

Figure 2: During training and fine-tuning, the norm of the neural network parameter continually increases, consistent with theory.

### 3.3 NEURAL PHYLOGENY DETECTION VIA CLUSTERING

In this section, we discuss a learning-free approach for neural phylogeny detection. The main idea is to view the neural phylogeny detection as a clustering problem, combining the detection of fine-tuning direction with traditional clustering methods.

Given a set of neural networks, the goal of neural phylogeny detection is to solve for an allocation matrix $\boldsymbol{A}$, which can be seen as a clustering task. Every model in $\{f^{(i)}\}_{i=1}^M$ can be regarded as a data point to be clustered. Matrix $\boldsymbol{A}$ allocates these data points into non-overlapping clusters. Each cluster should contain a parent model and the child models fine-tuned from it. For each cluster, the parent model should be identified as the cluster centroid.

However, the problem with previous clustering algorithms is that many algorithm not inherently include the identification of cluster centroids, such as DBSCAN (Hahsler et al., 2019). Even for those algorithms that can define cluster centroids during the clustering process, such as KMeans (Lloyd, 1982) and MeanShift(Wu & Yang, 2007), the cluster centroids identified by the algorithm may not necessarily be the parent model. Take the KMeans algorithm as an example. When the distance between points is measured using $\ell_2$ or $\ell_1$ norms, the centroid is the mean or median of the points within the cluster. However, from the analysis and empirical findings in Sec. 3.2, it can be seen that the parent model might not locate near the geometric center of the cluster. Because parent model has smaller parameter norm.

There are two approaches to integrating the identification of parent models with clustering algorithms. A straightforward method is, first, using a clustering algorithm to identify the clusters; then, for each cluster, identify the parent model in the cluster. Specifically, suppose $C \subset \{1, 2, \cdots, M\}$ is the set of indices of the models in one cluster, the index of the parent model in this cluster can be defined as

$$\arg\min_i ||\boldsymbol{\theta}^{(i)}||_2. \tag{2}$$

For the algorithms that require evaluation of cluster centroids during clustering, the other way of integrating the identification of parent models with the clustering algorithms is by incorporating the assessment of fine-tuning direction directly into the centroid evaluation process. Below we take the KMeans and MeanShift algorithms as examples to illustrate. In Appendix B.3 Fig. 5, using KMeans algorithm as an example, we provide a diagram of the algorithm's workflow to visually illustrate how our method integrates the finetuning direction identification into the original clustering algorithm.

**KMeans**. KMeans algorithm executes the assignment step and the update step iteratively. During the assignment step, each model is assigned to the nearest cluster. This step is essentially a neural lineage detection operation, which can be address effectively using the common distance measurements (Yu & Wang, 2024) and does not involves the identification of the parent model. Thus, we keep the original workflow in the assignment step and use the $\ell_2$ norm as the distance measurement to assign each model to the nearest cluster. In the update step, the KMeans algorithm updates the centroid of each cluster, which in our task should be the parent model. The prediction of the parent model in the cluster can be achieved using Eq. (2). However, during the iterative process, a single cluster may have none or more than one parent model. Therefore, to enhance the robustness of the method, we introduce a probability parameter $\alpha$. When updating the cluster centroid, with a probability of $\alpha$, the cluster's mean is used as the new centroid, and with a probability of $1 - \alpha$, the new centroid is chosen using Eq. (2). After the convergence of the algorithm, the final prediction of the parent model is drawn again using Eq. (2).

**MeanShift with a flat kernel**. Given a to-be-optimized cluster centroid, mean shift updates the centroid based on the mean of the points within the neighborhood of the centroid. The method for calculating the new cluster centroid can also be represented as finding the point that minimizes the sum of distances from the centroid to other points in the neighborhood, similar to the update step in the KMeans method. Thus, we employed the same approach as in KMeans by introducing a probability parameter $\alpha$ and randomly choose to use either the mean of the points in the neighborhood as the new centroid or use Eq. (2) to select the new centroid.

For approaches that identify the parent along the clustering, the choice of parameter $\alpha$ significantly influences the performance by altering the proportion of iterations using the predicted parent as the cluster centroid. We empirically studied the impact of $\alpha$ values on the final prediction accuracy in Fig. 3a. When $\alpha = 0$, each update uses the predicted parent as the cluster centroid. When $\alpha = 1$, the "identify

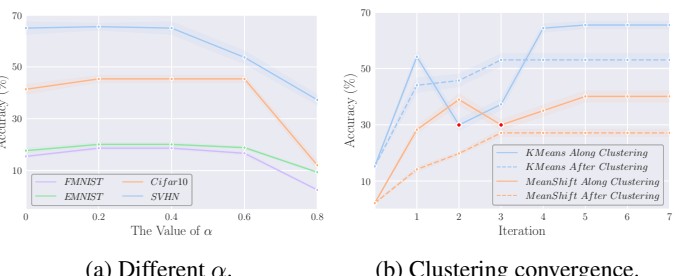

(a) Different $\alpha$.  (b) Clustering convergence.

Figure 3: Ablation on the value of $\alpha$ and the convergence of the clustering algorithms.

along" approach degenerates into the original clustering algorithm, always using the internal mean as the new centroid. However, neither of these extreme methods achieves optimal performance. Generally, using a relatively small $\alpha$, primarily relying on the predicted parent as the new centroid while occasionally using the mean, yields the best results.

Besides prediction accuracy, the convergence speed of the clustering algorithm is also a crucial factor in evaluation. We observe that in a clustering task involving 277 models, shown in Fig. 3b, both the original clustering algorithm and the one embedded with parent model identification typically converge within 4-5 iterations, demonstrating high efficiency. Additionally, we observe that some dig points appear in the "identify along" approach, marked by red dots ● in the figure. The occurrence of these dig points is due to the change in the centroid updating method, leading to fluctuations in node

allocation and consequent accuracy decrease. However, altering the method of centroid updating also corrects accumulated errors during iteration, resulting in higher accuracy upon full convergence.

### 3.4 NEURAL PHYLOGENY DETECTOR

In this section, we propose a learning-based phylogeny detector. We treat the task of phylogeny detection as a structure learning problem for a directed cyclic graph $G$. Given a set of neural networks $\{f^{(i)}\}_{i=1}^{M}$, each network is a node in graph $G$, with corresponding network parameter $\boldsymbol{\theta}^{(i)}$ being the feature of the node. The adjacency matrix $\boldsymbol{A}$ of $G$ is what we aim to predict in phylogeny detection. The structure of the phylogeny detector, as shown in Fig. 4, includes an edge encoder that takes the source node feature and the target node feature of each edge as input, outputs the latent embedding of each edge, and a transformer detector that predicts the probability

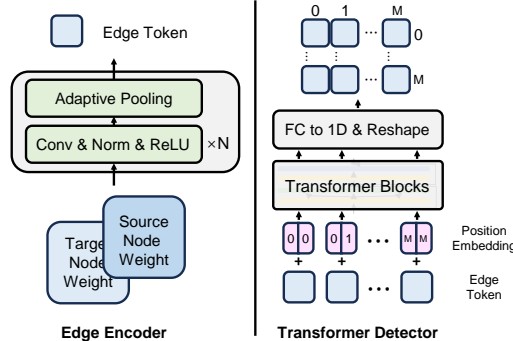

Figure 4: The proposed phylogeny detector.

of the existence of each edge. The following outlines the detailed workflow of the phylogeny detector.

First, we vectorize and reshape each weight into a matrix $\boldsymbol{\theta} \in \mathcal{R}^{H \times W}$, with shape depending on the specific architecture of the parent and child models. We stack the weights from node $i$ and node $j$ together and use it as the input feature of the edge $e_{i,j}$ and denote it by $\boldsymbol{\theta}^{(i,j)} \triangleq [\boldsymbol{\theta}^{(i)}; \boldsymbol{\theta}^{(j)}] \in \mathcal{R}^{2 \times H \times W}$. Such a stacking strategy is more effective than encoding the weights of the parent and child models independently, by allowing for element-wise comparison of the naturally aligned weights from different models, capturing subtle changes in weights caused by fine-tuning. We use $\boldsymbol{w} \in \mathcal{R}^{(M \times M) \times 2 \times H \times W}$ to denote the complete input of the edge encoder, which includes all edge features. We use a convolutional network as the edge encoder $\Phi$, applying batch normalization and ReLU activation after each convolution layer. At the end of $\Phi$, we use an adaptive pooling with an output size of $1 \times 1$ to transform each edge's latent embedding into a vector, which serves as the token input to the subsequent transformer detector $\Psi$. The output of $\Phi$ can be written as $\boldsymbol{w}^{enc} \triangleq \Phi(\boldsymbol{w}) \in \mathcal{R}^{(M \times M) \times d}$, where $d$ is the token size for the transformer detector.

We use position embedding $\boldsymbol{E}^{pos} \in \mathcal{R}^{(M \times M) \times d}$ to encode the indices of the source and target nodes of each edge, allowing the transformer detector to utilize the relative relationships between edges during processing. First, based on the node indices, we generate the position embedding for each node $\boldsymbol{e}^{pos} \in \mathcal{R}^{M \times (d/2)}$. The position embedding for the edge $e_{i,j}$ is the concatenation of the position embeddings of nodes $i$ and $j$, that is, $[\boldsymbol{e}_{i:}^{pos}, \boldsymbol{e}_{j:}^{pos}] \in \mathcal{R}^{d}$. The transformer's input is the sum of the edge tokens and their corresponding position embeddings. Finally, we use a linear head to transform the transformer's output for each edge token into a scalar score, which can be represented as

$$\boldsymbol{w}^{final} = \Psi(\boldsymbol{w}^{enc} + \boldsymbol{E}^{pos}), \ \boldsymbol{s} = Head(\boldsymbol{w}^{final}), \ \boldsymbol{w}^{final} \in \mathcal{R}^{(M \times M) \times d}, \text{ and } \boldsymbol{s} \in \mathcal{R}^{M \times M}, \quad (3)$$

where $\boldsymbol{s}$ is reshaped to ease the notation.

A characteristic of the directed graph formed by neural phylogeny is that there is at most one outgoing edge from each node. This results in the directed graph being very sparse, and using $\ell_2$ or binary cross-entropy loss on all edges would lead to imbalanced supervision signals. Therefore, we use cross-entropy as the loss function. For node $i$, if it is a child model, its ground truth label $gt_i$ is the index of its parent model; if it is a parent model, its ground truth label is defined as its own index $i$. The predicted probability of the edge from node $i$ to node $j$ is defined as $\boldsymbol{p}_{i,j} \triangleq \mathrm{softmax}(\boldsymbol{s}_{i,:})_j$. According, during inference, let $j^* \triangleq \arg\max_j p_{i,j}$, if $i = j^*$, we predict that model $i$ is a parent model; otherwise, we predict that model $i$ is fine-tuned from model $j^*$.

## 4 EXPERIMENTS

**Proof of Concept.** We experiment on 4 model architectures: a three-layer fully-connected network (FC Net) with ReLU activation and no normalization; a three-layer convolutional network (Conv

Table 2: Phylogeny detection performance for the vision models. The best (the second-best) ones are marked in **bold** (underlined).

| | Method | FMNIST | EMNIST | Cifar10 | SVHN | Cifar10 5shot | Cifar10 50shot | Cifar10 Imbalanced |
|---|---|---|---|---|---|---|---|---|
| FC Net | DBSCAN | 15.64±0.17 | 13.16±0.43 | 24.86±1.96 | 31.32±2.46 | 7.35±0.76 | 41.61±1.96 | 39.10±0.92 |
| | GMM | 9.48±0.36 | 3.43±0.16 | 65.71±1.59 | 46.59±1.19 | 9.71±0.84 | 42.34±1.85 | 65.93±1.12 |
| | KMeans | 15.80±0.39 | 12.54±0.47 | 53.11±2.40 | 45.38±1.25 | 9.71±0.84 | 43.80±2.88 | 61.54±0.81 |
| | MeanShift | 14.60±0.55 | 12.77±0.44 | 27.12±1.22 | 33.73±2.40 | 6.56±0.97 | 42.54±1.65 | 40.11±1.12 |
| | KMeans-$along$ | 18.64±0.49 | 20.09±0.56 | 65.53±1.26 | 45.38±1.15 | 9.81±0.54 | 88.32±0.70 | 62.24±0.61 |
| | MeanShift-$along$ | 15.48±0.21 | 13.55±0.45 | 40.11±1.89 | 34.13±2.49 | 12.60±0.33 | 98.87±0.02 | 56.59±1.43 |
| | Phylogeny Detector | **87.78±0.25** | **84.41±0.26** | **70.86±2.19** | **50.11±2.11** | **48.61±0.76** | **99.52±0.12** | **68.73±4.25** |
| ConvNet | DBSCAN | 28.30±0.85 | 45.83±1.20 | 27.60±1.22 | 44.53±0.57 | 12.32±0.83 | 43.10±2.36 | 42.89±2.28 |
| | GMM | 28.12±0.90 | 30.27±0.70 | 46.29±1.29 | 39.82±0.87 | 15.19±1.24 | 82.79±1.42 | 54.47±0.74 |
| | KMeans | 25.79±0.81 | 28.29±0.65 | 45.70±1.52 | 39.05±0.16 | 16.33±1.17 | 77.42±0.99 | 51.63±0.85 |
| | MeanShift | 18.15±0.84 | 15.52±0.32 | 14.24±0.79 | 21.39±0.70 | 6.20±0.73 | 94.62±1.23 | 24.19±1.32 |
| | KMeans-$along$ | 27.40±0.49 | 32.36±0.68 | 45.75±1.59 | 41.21±0.58 | 16.76±0.39 | 78.51±4.34 | 54.68±0.87 |
| | MeanShift-$along$ | 26.33±0.77 | 44.28±1.10 | 45.99±1.82 | 39.82±0.19 | 37.25±1.69 | 94.62±1.23 | 47.16±0.80 |
| | Phylogeny Detector | **83.64±0.77** | **60.12±1.56** | **99.17±0.29** | **98.86±0.42** | **82.54±0.77** | **99.81±0.15** | **96.97±0.98** |
| ResNet | DBSCAN | 19.43±1.46 | 38.50±0.92 | 43.70±2.64 | 55.17±2.80 | 11.95±0.61 | 28.14±3.55 | 34.69±2.00 |
| | GMM | 14.29±0.45 | 28.74±0.65 | 24.28±1.44 | 48.27±2.46 | 11.80±0.67 | 28.57±3.52 | 25.51±2.28 |
| | KMeans | 28.01±1.59 | 41.38±0.94 | 48.55±2.24 | 71.55±0.49 | 11.56±0.64 | 28.98±3.60 | 32.67±2.01 |
| | MeanShift | 18.43±1.56 | 37.36±0.98 | 43.75±2.74 | 55.23±2.48 | 11.60±0.65 | 28.84±3.52 | 32.41±1.95 |
| | KMeans-$along$ | 28.57±1.75 | 43.10±1.92 | 84.47±0.84 | 71.68±2.16 | 11.74±0.59 | 29.04±3.33 | 33.25±2.78 |
| | MeanShift-$along$ | 19.43±1.26 | 38.50±0.92 | 67.97±1.82 | 55.64±2.40 | 11.82±0.49 | 29.35±3.24 | 33.68±2.79 |
| | Phylogeny Detector | **83.01±0.64** | **88.46±1.57** | **90.18±0.77** | **88.39±0.73** | **53.99±1.26** | **57.94±1.35** | **87.01±1.30** |
| ViT | DBSCAN | 39.14±2.06 | 24.64±1.53 | 55.42±4.24 | 36.76±1.93 | 70.01±4.68 | 55.01±1.97 | 26.46±2.35 |
| | GMM | 47.83±2.79 | 21.75±1.71 | 72.80±3.38 | 47.60±2.55 | 55.01±0.02 | 53.36±2.52 | 25.01±1.62 |
| | KMeans | 44.94±2.52 | 25.38±1.54 | 70.61±3.83 | 46.33±2.34 | 50.17±0.05 | 56.21±2.51 | 25.79±1.63 |
| | MeanShift | 43.49±2.42 | 25.31±1.60 | 47.09±4.13 | 30.87±2.26 | 65.75±4.76 | 45.01±8.29 | 25.51±1.66 |
| | KMeans-$along$ | 46.38±2.00 | 69.57±4.15 | 70.71±3.39 | 70.61±2.96 | 80.01±4.68 | 80.01±4.68 | 67.27±2.53 |
| | MeanShift-$along$ | 47.83±2.79 | 72.47±3.97 | 73.55±3.03 | 55.89±3.15 | 70.01±6.12 | 45.53±4.28 | 54.41±1.79 |
| | Phylogeny Detector | **68.33±1.41** | **72.51±1.26** | **89.01±0.86** | **72.01±1.23** | 52.89±0.86 | 51.82±2.02 | **67.33±1.13** |

Net) with ReLU activation and no normalization; ResNet18; and ViT-Tiny. For fully-connected and convolutional networks, the parent models are trained by us on MNIST (Lecun et al., 1998) and CIFAR-100 (Krizhevsky et al., 2009). For ResNet and ViT, the parent models are collected from the timm (Wightman, 2019). Parent models are finetuned on FMNIST (Xiao et al., 2017), EMNIST-Letters (Cohen et al., 2017), and CIFAR10 (Krizhevsky et al., 2009) datasets to construct the child model sets. The average number of child(parent) models for the four architectures was 693(64), 756(44), 106(7), and 64(3), respectively. We compare the detection accuracy of methods in three categories: (1) DBSCAN (Hahsler et al., 2019), GMM (Ouyang et al., 2004), KMeans (Lloyd, 1982) and MeanShift (Wu & Yang, 2007) clustering algorithms with the "identify parent after clustering" strategy; (2) KMeans and MeanShift algorithms with the "identify parent along clustering" strategy. Thereafter, we denote them by KMeans-$along$ and MeanShift-$along$; and (3) the phylogeny detector. A prediction is considered correct only if the parent-child pair is accurately identified and the direction of fine-tuning is correctly determined. For learning-free methods, we reported the average accuracy over 5-fold experiments. For phylogeny detector, we split the child models into training, validation, and testing sets in a 7:1:2 ratio and report the average accuracy over five runs on the testing set.

Results are summarized in Tab. 2, with key observations as follows: (1) Instead of using a fixed statistical metric, learning-based detector generally outperforms the learning-free methods by a significant margin. Compared to the best-performing learning-free method in each setting, the phylogeny detector's performance is, on average, 29.43% higher. (2) Among the learning-free methods, no single clustering algorithm consistently outperforms the others. This is mainly due to the sensitivity of clustering methods to data structure, making the choices of hyperparameter significantly affects the performance of the clustering algorithm. (3) By incorporating parent model

Table 3: Cross-generational phylogeny detection.

|  |  | G2 | G3 | G4 |
|---|---|---|---|---|
| G1 | DBSCAN | 38.50±0.92 | 38.23±1.25 | 38.86±1.95 |
|  | GMM | 28.74±0.65 | 23.27±2.73 | 14.29±0.45 |
|  | KMeans | 41.38±0.94 | 23.85±2.56 | 14.35±0.52 |
|  | MeanShift | 37.36±0.98 | 37.35±1.05 | 37.27±2.01 |
|  | KMeans-*along* | 43.10±1.92 | 42.51±1.20 | 42.86±1.88 |
|  | MeanShift-*along* | 38.50±0.92 | 38.82±1.15 | 38.47±1.33 |
|  | Phylogeny Detector | **88.46±1.57** | **87.82±0.59** | **73.72±1.96** |
| G2 | DBSCAN | - | 40.35±0.62 | 38.86±1.95 |
|  | GMM | - | 29.24±1.32 | 14.86±0.53 |
|  | KMeans | - | 28.07±1.13 | 26.29±0.70 |
|  | MeanShift | - | 40.35±0.62 | 40.02±2.21 |
|  | KMeans-*along* | - | 42.69±1.04 | 40.86±1.05 |
|  | MeanShift-*along* | - | 40.96±0.65 | 40.57±2.28 |
|  | Phylogeny Detector | - | **77.56±0.74** | **72.75±1.35** |
| G3 | DBSCAN | - | - | 68.01±2.09 |
|  | GMM | - | - | 67.43±0.35 |
|  | KMeans | - | - | 53.71±1.59 |
|  | MeanShift | - | - | 58.86±2.10 |
|  | KMeans-*along* | - | - | 56.57±1.53 |
|  | MeanShift-*along* | - | - | 68.05±2.12 |
|  | Phylogeny Detector | - | - | **88.46±0.92** |

Table 4: Ablations.

| Embedding Dimension | FMNIST | EMNIST |
|---|---|---|
| 8 | 84.44 | 77.42 |
| 16 | 87.51 | 81.99 |
| 32 | 87.78 | 83.07 |
| 64 | **87.91** | **83.11** |

(a) Embedding dimension.

| Encoder Depth | FMNIST | EMNIST |
|---|---|---|
| 3 | 85.83 | 79.84 |
| 5 | **88.89** | **84.41** |
| 8 | 88.33 | **84.41** |
| 10 | 86.39 | 83.87 |

(b) Encoder depth.

| Transformer Depth | FMNIST | EMNIST |
|---|---|---|
| 0 | 87.51 | 80.64 |
| 1 | **87.78** | 81.99 |
| 2 | 85.83 | **83.07** |
| 3 | 84.72 | 78.76 |

(c) Transformer depth.

identification into the clustering process, the performance of KMeans-*along* and MeanShift-*along* algorithms outperforms the original KMeans and MeanShift. The performance of KMeans and MeanShift methods improved by an average of 9.95% after using the "identify along" strategy.

**Few-Shot and Imbalanced Data.** We further evaluated the methods on CIFAR10 datasets under few-shot and imbalanced[1] learning settings in Tab. 2. Generally, the learning-based method still performs the best, and the strategy of identifying parents along clustering improves the performance of KMeans and MeanShift algorithms. However, in the few-shot setting for ViT models, due to the difficulty of fine-tuning, we obtained only an average of 16 child models, resulting in insufficient training samples for the phylogeny detector. Consequently, the learning-based method perform worse than the learning-free methods in this specific case.

**Cross-Generational Phylogeny.** We define the 7 parent ResNet18 models as the First Generation (G1) models and perform three rounds of fine-tuning, one after another, on the EMNIST (G2), FMNIST (G3), and EMNIST-Balanced (G4) datasets. Tab. 3 compares the cross-generational phylogeny detection performance. The learning-based method and the strategy of identifying parents along clustering generally maintain their advantages. In the table, we use pink squares ■ to indicate the relative performance of each method under different settings, with darker colors representing higher accuracy and lighter colors representing lower accuracy. It can be seen that as the generational gap increases, the performance of all methods generally declines.

**Ablation on Detector Architecture.** We conduct ablation on the structure of the phylogeny detector to observe changes in accuracy with increasing network complexity. We examine, in Tab. 4, three factors: the embedding dimension of the transformer, the number of layers in the edge encoder, and the number of layers in the transformer. The main observation was that due to the limited size of training samples, accuracy does not consistently improve with increased network complexity. The combination of a 5-layer encoder and a 1-layer transformer is generally optimal.

**Experiments on Open-Sourced Stable Diffusion and Llama.** We conducted experiments on more complex Diffusion Models and Llama Models. Specifically, we collected 366 Stable Diffusion models and 95 Llama models from the HuggingFace. Compared to the models used in the primary experiments, these models have larger parameter sizes, more intricate architectures, and more diverse training configurations. The child models were trained on different datasets, using various hyperparameters and training techniques—not limited to full-parameter fine-tuning but also including

---

[1]created by down-sampling the first five categories to its 1/10.

Table 5: The performance of the learning-free method on open-source Stable Diffusion and Llama models, as well as on Stable Diffusion models fine-tuned using DreamBooth.

| | Identify after Clustering | | | | Identify along Clustering | | | |
| | KMeans | | MeanShift | | KMeans | | MeanShift | |
| | Avreage | Best | Avreage | Best | Avreage | Best | Avreage | Best |
|---|---|---|---|---|---|---|---|---|
| Stable Diffusion | 14.92±7.45 | 18.64 | 18.97±1.01 | 19.02 | 78.57±21.28 | 99.73 | 68.16±24.75 | 80.54 |
| Llama | 0±0 | 0 | 0±0 | 0 | 48.83±48.37 | 97.67 | 75.58±37.81 | 95.34 |
| DreamBooth | 100.0±0 | 100.0 | 100.0±0 | 100.0 | 100.0±0 | 100.0 | 100.0±0 | 100.0 |

Table 6: The performance of the learning-based method on open-source Stable Diffusion and Llama models, as well as on Stable Diffusion models fine-tuned using DreamBooth.

| | SD 1.4 | SD 2 | SD 2.1 | Llama 2 | Llama 3 |
|---|---|---|---|---|---|
| Overall Accuracy | 99.44±0.25 | 99.12±0.74 | 99.33±0.47 | 92.86±0.70 | 88.37±0.60 |
| Lineage Accuracy | 100.0±0.00 | 100.0±0.00 | 100.0±0.00 | 100.0±0.00 | 100.0±0.00 |
| Direction Accuracy | 98.64±0.21 | 96.55±0.63 | 97.72±0.27 | 91.81±0.40 | 80.00±0.44 |

techniques like LoRA fine-tuning. Subsequently, using DreamBooth (Ruiz et al., 2023), we fine-tuned Stable Diffusion 1.4 and Stable Diffusion 2.1 on the official DreamBooth dataset, resulting in 62 models (30 derived from each parent model and two parent models).

We evaluated the performance of our learning-free technique on these models, with results presented in Tab. 5. Consistent with earlier findings, the performance of the learning-free approach significantly improved using the "Identify along Clustering" strategy. However, we observed that the performance of the learning-free method on these models was unstable, with a substantial gap between its best and average performance. We further tested our learning-based method on open-sourced models collected from HuggingFace. The experimental results are reported in Tab. 6, where we evaluated three metrics: Overall Accuracy, Lineage Accuracy, and Direction Accuracy. Let $N$ denote the total number of models, then, there are $N_{pair} = \frac{N(N-1)}{2}$ model pairs. For the $i$-th model pair, $gt_i$ is the ground truth label, which can take three possible values: $\{0 = \text{No Finetuning Relationship}, 1 = \text{The First Model is Parent}, 2 = \text{The Second Model is Parent}\}$. $pred_i$ is the predicted label for the $i$-th pair. The definitions of these metrics are as follows:

$$\text{Overall Accuracy} = \frac{1}{N_{pair}} \sum_{i=1}^{N_{pair}} \mathbf{1}(gt_i = pred_i),$$

$$\text{Lineage Accuracy} = \frac{1}{N_{pair}} \sum_{i=1}^{N_{pair}} \mathbf{1}((gt_i = pred_i = 0) \text{ OR } (pred_i \neq 0 \text{ AND } gt_i \neq 0))$$

$$\text{Direction Accuracy} = \frac{1}{\sum_{i=1}^{N_{pair}} \mathbf{1}(gt_i \neq 0)} \sum_{i=1}^{N_{pair}} \mathbf{1}(gt_i \neq 0 \text{ AND } gt_i = pred_i).$$

The results in the Tab. 6 demonstrate that the learning-based approach significantly outperformed the learning-free method, achieving nearly 100% accuracy even on complex open-sourced models.

## 5 CONCLUSION

We investigate the novel task of neural phylogeny detection, which, given a set of neural network models, aims to identify parent-child model pairs connected through the fine-tuning behavior and determine the direction of fine-tuning. By treating neural phylogeny detection as a clustering task or a graph structure learning task, we propose both learning-free and learning-based methods. Extensive experiments demonstrate the effectiveness of our proposed methods across various network architectures and learning settings, and for cross-generational phylogeny detection.

## ACKNOWLEDGEMENT

This project is supported by the National Research Foundation, Singapore, and Cyber Security Agency of Singapore under its National Cybersecurity R&D Programme and CyberSG R&D Cyber Research Programme Office (Award: CRPO-GC1-NTU-002).

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

# Neural Phylogeny
## *- Supplementary Material -*

## A   EXPERIMENTS WITH STABLE DIFFUSION AND LLAMA

### A.1   COLLECTION OF OPEN-SOURCE MODELS

For the open-source Stable Diffusion and Llama models, we downloaded them directly from Hugging-Face. We selected five parent models in total, "CompVis/stable-diffusion-v1-4", "stabilityai/stable-diffusion-2", "stabilityai/stable-diffusion-2-1", "meta-llama/Llama-2-7b-hf" and "meta-llama/Meta-Llama-3-8B". Hugging Face provides a list of fine-tuned child models for each parent, and we sorted these lists by "trending" and downloaded the top models. For the model "CompVis/stable-diffusion-v1-4", we attempted to download the top 200 child models. For "stabilityai/stable-diffusion-2", which has 111 child models, we attempted to download the top 100. For "stabilityai/stable-diffusion-2-1", with 152 child models, we attempted to download the top 150. As for the LLaMA models "meta-llama/Llama-2-7b-hf" and "meta-llama/Meta-Llama-3-8B", due to their large sizes (14-16GB), we attempted to download the top 50 child models for each.

We excluded models that could not be downloaded, were mislabeled, or were unusable as standalone models. For example, some 70B models were incorrectly labeled as fine-tuned from 7B models, or certain LoRA-fine-tuned models only released the LoRA parameters, making them incompatible with a unified interface. After these exclusions, we collected 191 child models for "CompVis/stable-diffusion-v1-4", 68 for "stabilityai/stable-diffusion-2", 107 for "stabilityai/stable-diffusion-2-1", 48 for "meta-llama/Llama-2-7b-hf", and 47 for "meta-llama/Meta-Llama-3-8B".

### A.2   COLLECTION OF DREAMBOOTH-FINETUNED MODELS

We use the "diffusers" library to implement the DreamBooth fine-tuning and image generation for diffusion models. The three diffusion models we used are "CompVis/stable-diffusion-v1-4" and "stabilityai/stable-diffusion-2-1". The fine-tuning dataset is the official DreamBooth release dataset. We fine-tune on a all of 30 sets of images, resulting in 60 child diffusion models.

### A.3   EXPERIMENTS WITH OPEN-SOURCED MODELS IN FIGURE 1

In this experiment, we focused on detecting the existence and direction of the fine-tuning relationship between two models. For example, given Stable Diffusion 1.4 and another model $f$, the detector determines whether a finetuning relationship exists between Stable Diffusion 1.4 and $f$, and if so, the detector further identifies which model is the parent. This is a practical scenario, as in some cases, we want to determine whether a specific model is fine-tuned from a particular open-source model, or we want to detect the fine-tuning relationship between two models without interference from other models. To achieve this, we trained a detector. The network architecture of the detector is consistent with the one described in Fig. 4, but since only two models are involved, we replaced the transformer block with an MLP-based prediction head. The prediction head is designed to output three categories: no fine-tuning relationship between the two models, the first model is the parent of the second, and the second model is the parent of the first.

We randomly split the collected child models into training, validation, and test subsets in a 7:1:2 ratio. The models in each subset were paired with the parent models to form the input for the detector. The experiments for Stable Diffusion and LLaMA models were conducted separately. That is, in the Stable Diffusion experiment, possible combinations included (Stable Diffusion 1.4, a child of Stable Diffusion 1.4), (Stable Diffusion 1.4, a child of Stable Diffusion 2), (Stable Diffusion 1.4, a child of Stable Diffusion 2.1), (a child of Stable Diffusion 1.4, Stable Diffusion 1.4), (a child of Stable Diffusion 2, Stable Diffusion 1.4), (a child of Stable Diffusion 2.1, Stable Diffusion 1.4) and other combinations sharing similar patterns. Other training details are consistent with the experiments in Tab. 2.

We recorded the detector's accuracy for detecting the existence of the finetuning, accuracy for detecting the direction of the finetuning and the overall accuracy. The reported "Lineage Acc" is defined as the percentage of cases where the existence of a finetuning relationship between the input model pair is correctly detected. The reported "Direction Acc" is defined as the percentage of cases where the parent model in a detected finetuning relationship is correctly identified. The "Overall Acc" is defined as the percentage of cases where both the existence of a finetuning relationship and the parent model are correctly identified. All reported results are computed from the testing set. The

results are presented in Fig 1 and Tab. 6. Our method achieves near 100% detection accuracy. For each parent model, we also display some of its child models along with the detector's predicted probabilities that they were finetuned from the parent model in Fig 1.

### A.4 Performance of the Learning-Free Detection Method with Open-Source Models

We also tested the performance of the learning-free methods on the collected open-source Stable Diffusion and Llama models. The input parameters used were the same as in the experiments of the learning-based method, but here the goal was to detect all fine-tuning relationships and directions among the models in one pass. The experiments for Stable Diffusion and Llama models were conducted separately. Each experiment was repeated 10 times, and we recorded the average accuracy and the best accuracy across the trials. The specific results are shown in the first two rows of Tab. 5.

In terms of the best achievable performance, the learning-free clustering method, which incorporates the idea of "distinguish along clustering", achieved significantly better predictions compared to the original KMeans and MeanShift methods. However, the learning-free method lacked stability and was highly sensitive to the initialization. Across multiple runs, many failed attempts were observed, leading to a low average accuracy and a high variance.

### A.5 Experiments with DreamBooth-Finetuned Models in Figure 1

We conduct phylogeny detection experiments using four learning-free strategies. The detailed results are shown in the last row of Tab. 5, which are also illustrated in the bottom right part in Figure 1. For the clustering algorithm's input, we used the "down_blocks.0.attentions.0.transformer_blocks.0.ff.net.2.bias" in the diffusion model's UNet encoder. Our method achieves a high accuracy in detecting the fine-tuning relationships and directions among 62 models in one pass.

## B Other Additional Results and Illustrations

### B.1 Experimental Results on Pet and DTD

Table 7: Phylogeny detection performance for the vision models on the Pet and DTD datasets. The best (the second-best) ones are marked in **bold** (underlined).

| Model | Method | | DTD | Pet |
|---|---|---|---|---|
| ResNet | Identify after Clustering | DBSCAN | 22.93±2.70 | 46.88±3.74 |
| | | GMM | 22.86±2.63 | 38.29±3.03 |
| | | KMeans | 23.45±3.01 | 46.88±3.75 |
| | | MeanShift | 22.74±2.54 | 46.52±3.66 |
| | Identify along Clustering | KMeans | 41.69± 3.90 | 47.42±3.74 |
| | | MeanShift | 41.54±4.09 | 47.36±3.54 |
| | Phylogeny Detector | | **50.44±1.85** | **49.61±1.83** |
| ViT | Identify after Clustering | DBSCAN | 71.87±4.76 | 2.08±1.04 |
| | | GMM | 59.38±5.40 | 2.25±1.13 |
| | | KMeans | 68.74±4.69 | 2.80±1.40 |
| | | MeanShift | 72.52±4.81 | 2.17±1.09 |
| | Identify along Clustering | KMeans | 68.86±3.30 | 3.07±1.12 |
| | | MeanShift | 72.87±4.76 | 2.91±1.18 |
| | Phylogeny Detector | | **82.54±1.16** | **50.79±1.07** |

The experimental results on the Pet and DTD datasets, shown in Tab. 7, are consistent with findings on other datasets. The learning-based method still outperforms the learning-free methods. Among the

Table 8: Performance of the learning-based detector under across-architecture generalization.

|  | VIiT | | Conv Net | |
|---|---|---|---|---|
|  | Cifar10 | SVHN | Cifar10 | SVHN |
| Best Learning-Free Method | 73.55% | 70.61% | 46.29% | 44.53% |
| Without Generalization | 89.01% | 72.01% | 99.17% | 99.86% |
| With Generalization | 62.27% | 60.67% | 83.33% | 74.62% |

various learning-free clustering methods, it is difficult to determine an absolute best. However, incorporating parent model identification into the clustering process generally improves the performance of the clustering algorithms.

## B.2 GENERALIZABILITY OF LEARNING-BASED DETECTOR

In the main experiments, we controlled the training and testing sets of the learning-based detector to have different child models (Appendix C.3). This implicitly requires the detector to be able to generalize to unseen child models. Such experimental setup is based on the perspective of pretrained model owners. They possess parent models and some child models and aim to detect from which pretrained model external child models are finetuned.

We further explored the detector's ability to generalize across different architectures. We trained the detector using FC-Net and tested it on VIT and ConvNet architectures. The results are shown in Tab. 8. In the table, the row "Best Learning-Free Method" refers to the highest accuracy of learning-free methods in the main result. Learning-free methods do not require training and thus do not face generalization issues. The row labeled "Without Generalization" corresponds to the accuracy when no cross-architecture generalization is required. These models are trained and tested on the same architecture-dataset pair. The row "With Generalization" corresponds to the scenario where the detector is trained on FC-Net and directly tested on VIT and ConvNet architectures. As shown, the learning-based detector demonstrates acceptable cross-architecture generalizability. Even when generalization to entirely new neural network architectures is required, the detector still achieves acceptable accuracy. Notably, in the ConvNet experiments, the accuracy of the directly generalized detector remains significantly higher than that of learning-free methods.

## B.3 DIAGRAM ILLUSTRATIONS ON THE KMEANS-ALONG METHOD

In Fig. 5, we illustrate the KMeans-along algorithm. The KMeans-along algorithm follows the same procedure as the original KMeans algorithm during the assignment step. As shown on the left part of Fig. 5, each point is assigned to the cluster whose centroid is closest to it based on distance. The key difference between KMeans-along and the original KMeans algorithm lies in the update step, illustrated on the right part of Fig. 5. In this step, the centroid of each cluster is updated. When $\alpha = 1$, the KMeans-along algorithm reduces to the original KMeans algorithm, as shown in the upper-right part of Fig. 5, using the geometric center as the centroid for each cluster. However, when $\alpha < 1$, KMeans-along updates the centroid by selecting the point most likely to belong to the parent model with a probability of $1 - \alpha$, as shown in the lower-right part of Fig. 5.

## C OTHER IMPLEMENTATION DETAILS

Experiments are conducted on one Nvidia RTX 4090. For the parent and child models used, we followed the method of collecting parent and child models from previous work (Yu & Wang, 2024). However, compared to previous work, we collected more models with a greater variety of structures and in larger quantities. In this work, we consider finetuning all the parameters.

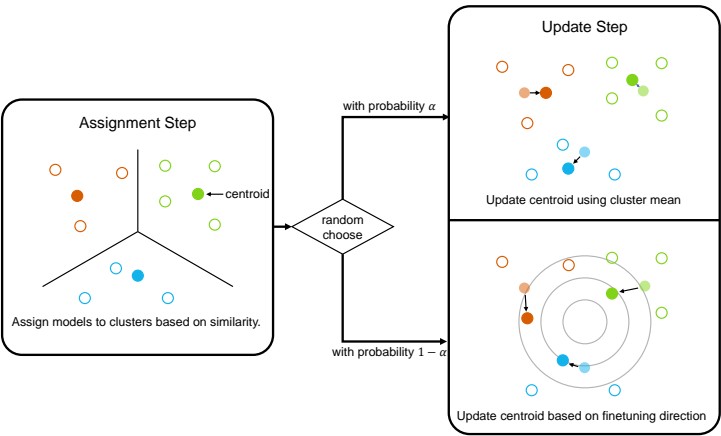

Figure 5: Illustrative diagram for the process of KMeans-along method.

## C.1 EXPERIMENTS IN TABLE 2 AND TABLE 5

### C.1.1 MODEL SET

We conducted experiments on four types of neural network architectures: a three-layer fully-connected network with ReLU activation; a three-layer convolutional network with ReLU activation; ResNet18; and ViT-Tiny.

The hidden layers of the fully-connected network are fixed at the sizes of 1024, 256, and 128. The convolutional network use convolutional kernels of size 3, followed by two fully-connected layers with latent feature dimensions set to 128. The input for the convolutional network is fixed at 28x28; images not meeting this size are resized accordingly. The output dimensions for both networks are adjusted based on the dataset. For the fully-connected and convolutional networks, the parent and child models are trained by ourselves. We first train fully-connected and convolutional networks on the MNIST and CIFAR-100 datasets, creating four parent model categories: fully-connected + MNIST, fully-connected + CIFAR-100, Convolutional + MNIST, Convolutional + CIFAR-100. The training process use a learning rate of either 0.01 or 0.001, a batch size of either 256 or 1024, and parameter initialization with either Kaiming Uniform or Kaiming Normal, using eight random seeds. This results in 64 training configurations. The number of training epochs is set to 50, and Adam optimizer is used. For the MNIST dataset, we only select models with an accuracy greater than 80% as parent models. For the CIFAR-100 dataset, models with an accuracy greater than 50% are selected as parent models. Finally, we obtain 64 fully-connected models on MNIST, 17 fully-connected models on CIFAR-100, 64 convolutional models on MNIST, and 44 convolutional models on CIFAR-100. Starting from these parent models, we fine-tune the child models. Models trained on MNIST are fine-tuned on FMNIST and EMNIST, while parent models from CIFAR-100 are fine-tuned on SVHN, CIFAR-10, and its variants. The fine-tuning learning rate is set to one of 0.01, 0.001, or 0.0001, with a batch size of either 256 or 1024, using four random seeds. The number of fine-tuning epochs is set to 30, using the Adam optimizer. We only select models with an accuracy greater than 50% as child models. Finally, the fully-connected network obtain 1202, 1220, 160, 232, 364, 264, and 260 models on FMNIST, EMNIST, CIFAR-10, SVHN, CIFAR-10-5shot, CIFAR-10-50shot, and CIFAR-10-Imbalanced, respectively. The convolutional network obtain 1050, 1100, 293, 826, 306, 168, and 448 models on FMNIST, EMNIST, CIFAR-10, SVHN, CIFAR-10-5shot, CIFAR-10-50shot, and CIFAR-10-Imbalanced, respectively. This model set is significantly larger than previous works.

For ResNet and ViT-Tiny models, we directly collect publicly released models from the timm library as parent models. The ResNet models used were "resnet18.a1_in1k", "resnet18.a2_in1k", "resnet18.a3_in1k", "resnet18.gluon_in1k", "resnet18.fb_ssl_yfcc100m_ft_in1k", "resnet18.fb_swsl_ig1b_ft_in1k", and "resnet18.tv_in1k", totaling seven models, all trained on ImageNet. The ViT-Tiny models used were "vit_tiny_r_s16_p8_224.augreg_in21k_ft_in1k", "vit_tiny_r_s16_p8_224.augreg_in21k", and "vit_tiny_patch16_224.augreg_in21k", totaling three models. We fine-tuned these parent models on all datasets shown in Tables 2 and 5. All input images are resized to 224x224. Grayscale images are duplicated three times to convert them into

three-channel images. The network's output dimensions are adjusted according to the dataset. The fine-tuning and model selection processes are consistent with those used for the fully-connected and convolutional networks. Finally, we obtain 168 child ResNet18 models on each dataset.we obtain 65, 65, 64, 64, 16, 16, 64, 44, and 60 child ViT-Tiny models on FMNIST, EMNIST, CIFAR-10, SVHN, CIFAR-10-5shot, CIFAR-10-50shot, CIFAR-10-Imbalanced, Pet, and DTD, respectively.

In our experiments, we combined the child models with their corresponding parent models to form the model sets for phylogeny detection.

### C.1.2   THE CHOICE OF WEIGHT

Using all the neural network parameters would result in significant complexity. In our experiments, we select only a subset of the neural network parameters for the phylogeny detection task. For fully-connected networks, we use the weight tensor of the first linear layer. For convolutional networks, we use the convolutional kernels of the first convolutional layer. For ResNet networks, we use the convolutional kernels of the first convolutional layer in the second ResNet block. For ViT, we used the weight tensor of the last linear transformation in the third transformer block.

### C.1.3   IMPLEMENTATION OF THE LEARNING-FREE METHODS

We use the DBSCAN and GMM implementations from the sklearn library. The search range for the eps parameter in DBSCAN is 0.5, 0.1, 1.0, 2.0, and 5.0. We implement the KMeans and Mean Shift algorithms ourselves. The search range for the bandwidth parameter in Mean Shift is 0.5, 1.0, 2.0, 3.0, and 5.0. For the identify parent along the clustering strategy, the search range for $\alpha$ is 0.0, 0.2, 0.4, 0.6, and 0.8. For algorithms that require the number of clusters as input, such as KMeans and GMM, we provide the actual number of parent models as the number of clusters. The maximum number of iterations for clustering is set to 50, but the algorithms often converge in fewer than 10 iterations.

### C.2   EXPERIMENTS IN TABLE 3

This experiment aimed to test the ability to perform phylogeny detection across generations. The method for constructing the model set to be detected followed the same approach as in previous work (Yu & Wang, 2024), using ResNet18 as the model architecture. The first generation consist of the original seven ResNet models. The second generation models are obtained by fine-tuning the first-generation models on the EMNIST-Letters dataset, using the child models collected in Table 2 experiments. From the second generation, we select seven models with highest accuracy as the parent models for training the third generation, and ensure that these models have different first-generation parent models. These seven models are fine-tuned on the FMNIST dataset to obtain the third-generation models. From the third-generation models, we again select seven models with the highest accuracy, ensure different first-generation ancestors, and fine-tune them on the EMNIST-Balanced dataset to obtain the fourth-generation models. The fine-tuning settings and model selection methods are consistent with those in Table 2 experiments. We obtain 168 models each for the third and fourth generations.

For the experiments involving the first and third generations, we combine the original seven ResNet models with the third-generation models fine-tuned on FMNIST to form the model set to be detected. For the first and fourth generations, we combine the original seven ResNet models with the fourth-generation models fine-tuned on EMNIST-Balanced. For the experiments between the second and third generations, we combine the seven models selected from the second generation fine-tuned on FMNIST with the third-generation models fine-tuned on FMNIST. The experiments between the third and fourth generations and between the second and fourth generations are constructed similarly.

The implementation of learning-free and learning-based methods and the hyperparameter search in these experiments are consistent with previous experiments.

### C.3   PHYLOGENY DETECTOR

Given a set of neural network models, similar to the learning-free method, we only extract part of the parameters from each model as the phylogeny detector input. The specific parameter selection

remains consistent with the learning-free method. This allows us to obtain features for all nodes in the directed graph, and we need to predict the edges in the graph. Similar to previous methods for node detection and edge detection on a single graph, we split the graph into several subgraphs, using a portion for training and the remaining parts for validation and testing.

Specifically, for each subgraph, we randomly select $P$ parents and $C$ child models for each parent, combining $P + P \times C$ models to form a subgraph. The values of $P$ and $C$ depend on the total number of parent and child models and the GPU memory limitations. For fully-connected, convolutional, and ResNet networks, $P$ is set to 4, and for ViT models, $P$ is set to 3. $C$ is always set to 6. The resulting subgraph sizes are 28 or 21. We divide the obtained subgraphs into training, validation, and test sets in a 7:1:2 ratio. Since the number of parent models is limited, we use it in the training, validation, and test sets, but we ensure that child models are unique to each set. If a child model is included in a training subgraph, it does not appear in the validation or test subgraphs. If a child model is included in a validation subgraph, it does not appear in the training or test subgraphs.

To increase the number of training samples, we use a strategy of repeated sampling of the parameters to create more samples. For example, in the experiment with fully-connected networks trained on MNIST and their child models, we use the weight matrix of the first linear layer, which has a size of 1024x784. Instead of setting the input size of the phylogeny edge encoder to 2x1024x784, we set it to 2x32x32. Thus, we used only part of the weight matrix, allowing us to sample multiple times from this matrix to create more than one samples. Specifically, once which models are included in a subgraph is determined, we sample 32x32 indices based on the shape of the weight matrix of the first linear layer. For each model included in the subgraph, we pick the elements from the weight matrix according to the sampled indices. Different subgraphs have different index sets. Given the models in a subgraph, we sample multiple times to create several data samples. We typically repeating the sampling 10 times. In the few-shot experiment with the ViT structure, due to the small number of models, we repeated the sampling 40 times. Even with this, we could not obtain a large enough dataset to improve the performance of the phylogeny detector in the ViT few-shot experiment.

Except in the architecture ablation experiment, we use a 5-layer edge encoder and a 1-layer transformer detector in all other experiments. The edge encoder's kernel size is set to 3, and the latent embedding size is set to 32. The training learning rate is 0.01, the batch size is 1, and the epoch is 100 for all experiments. The Adam optimizer is used.

### C.4 Experiment in Figure 2

We train eight ResNet18 models from scratch with a learning rate of 0.01 or 0.001, a batch size of 256 or 1024, using two random seeds. The training dataset is CIFAR-100, and the Adam optimizer is used, with the number of iterations set to 12,000. Subsequently, from each parent model, we fine-tune four child models with a learning rate of 0.01 or 0.001, using two random seeds. The fine-tuning dataset is CIFAR-10. The dark lines in the figure represent the average performance of the eight parent models and thirty-two child models. The lighter lines depict the trajectories of some specific parent and child models.

### C.5 Experiment in Figure 2 (Previous Version, Left Here only for Reviewers' Reference)

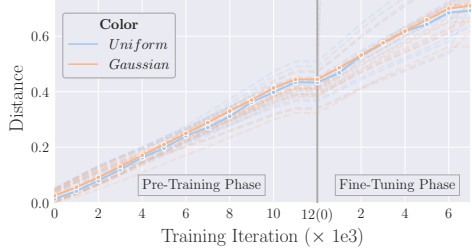

Figure 6: During training and fine-tuning, the distance between parameter and the fake initialization continually increases, consistent with theory.

We train eight ResNet18 models from scratch with a learning rate of 0.01 or 0.001, a batch size of 256 or 1024, using two random seeds. The training dataset is CIFAR-100, and the Adam optimizer is used, with the number of iterations set to 12,000. Subsequently, from each parent model, we fine-tune four child models with a learning rate of 0.01 or 0.001, using two random seeds. The fine-tuning dataset is CIFAR-10. The dark lines in the figure represent the average performance of the eight parent models and thirty-two child models. The lighter lines depict the trajectories of some specific parent and child models.

In our theory, the only restriction on the distribution of the fake initialization is that it has zero mean, with no constraints on the variance and the type of the distribution. We tested with two types of distributions and with various variances, including $\{1e-3, 1e-2, 1e-1, 1, 10\}$, and found that the resulting monotonic increasing trend remains stable. The plotted graphs are consistent with those in Figure 3. Note that, the variance of $1e-3$ is smaller than the variance when using Kaiming initialization, while a variance of 10 is much larger than the scale of the neural network parameters after training. Thus, this confirms that the choice of the distribution for $z$ is, as predicted by theory, highly robust.

## D  DERIVATIONS

### D.1  DERIVATION OF THE PROPOSITION

**Proposition 2.** *Let $f_t$ denote a neural network trained using gradient flow on the dataset $\mathcal{D} = (\mathcal{X}, \mathcal{Y})$ at time $t$. Let $\boldsymbol{\theta}_t \in \mathcal{R}^{d_{\boldsymbol{\theta}} \times 1}$ denote the vectorized parameters of $f_t$, where $d_{\boldsymbol{\theta}}$ is the number of parameters. Let $f_0$ denote the neural network at the initialization with initial parameters $\boldsymbol{\theta}_0$. Let $\eta$ denote the learning rate. Then, with MSE loss $\mathcal{L} \triangleq \frac{1}{2}\|f(\mathcal{X}) - \mathcal{Y}\|_2^2$, the Euclidean norm $\boldsymbol{\theta}_t$ can be written as,*

$$\frac{1}{d_{\boldsymbol{\theta}}}\|\nabla_{\boldsymbol{\theta}_0} f_0(\mathcal{X})^T \Theta^{-1}(I - e^{-\eta\Theta t})(f_0(\mathcal{X}) - \mathcal{Y})]\|_2^2 + \epsilon, \tag{4}$$

*which increases over time $t$.*

*Proof.* In the derivation, we use the dynamic of the linearized $f_t$ to approximate the dynamic of the original non-linear $f_t$. Such linearization has been demonstrated to be an effective approximation of the dynamic of the original $f_t$ with high accuracy (Lee et al., 2019b). To simplify the notation, we keep using the $f_t$ to denote linearized network. Given the input $\mathcal{X}$, the output of the linearized network is defined as $f_t(\mathcal{X}) = \triangleq f_0(\mathcal{X}) + \nabla_{\boldsymbol{\theta}_0} f_0(\mathcal{X})(\boldsymbol{\theta}_t - \boldsymbol{\theta}_0)$. Under the gradient flow, the dynamic of the parameters follows

$$\dot{\boldsymbol{\theta}}_t = -\eta \nabla_{\boldsymbol{\theta}_0} f_0(\mathcal{X})^T \nabla_{f_t(\mathcal{X})} \mathcal{L}. \tag{5}$$

Under MSE loss the solution of $\boldsymbol{\theta}_t$ is

$$\boldsymbol{\theta}_t = \boldsymbol{\theta}_0 - \nabla_{\boldsymbol{\theta}_0} f_0(\mathcal{X})^T \Theta^{-1}(I - e^{-\eta\Theta t})(f_0(\mathcal{X}) - \mathcal{Y}), \tag{6}$$

where $I$ is the identity matrix, and $\Theta \triangleq \nabla_{\boldsymbol{\theta}_0} f_0(\mathcal{X}) \nabla_{\boldsymbol{\theta}_0} f_0(\mathcal{X})^T$ is the tangent kernel at the initialization. Thus, the $\ell_2$ norm of the parameters can be written as

$$\frac{1}{d_{\boldsymbol{\theta}}}\|\boldsymbol{\theta}_t\|_2^2 = \frac{1}{d_{\boldsymbol{\theta}}}\|\boldsymbol{\theta}_0 - \nabla_{\boldsymbol{\theta}_0} f_0(\mathcal{X})^T \Theta^{-1}(I - e^{-\eta\Theta t})(f_0(\mathcal{X}) - \mathcal{Y})\|_2^2 \tag{7}$$

$$= \frac{1}{d_{\boldsymbol{\theta}}}[\boldsymbol{\theta}_0 - \nabla_{\boldsymbol{\theta}_0} f_0(\mathcal{X})^T \Theta^{-1}(I - e^{-\eta\Theta t})(f_0(\mathcal{X}) - \mathcal{Y})]^T$$
$$\times [\boldsymbol{\theta}_0 - \nabla_{\boldsymbol{\theta}_0} f_0(\mathcal{X})^T \Theta^{-1}(I - e^{-\eta\Theta t})(f_0(\mathcal{X}) - \mathcal{Y})] \tag{8}$$

$$= \frac{1}{d_{\boldsymbol{\theta}}}[\nabla_{\boldsymbol{\theta}_0} f_0(\mathcal{X})^T \Theta^{-1}(I - e^{-\eta\Theta t})(f_0(\mathcal{X}) - \mathcal{Y})]^T$$
$$\times [\nabla_{\boldsymbol{\theta}_0} f_0(\mathcal{X})^T \Theta^{-1}(I - e^{-\eta\Theta t})(f_0(\mathcal{X}) - \mathcal{Y})]$$
$$- \frac{2}{d_{\boldsymbol{\theta}}}[\nabla_{\boldsymbol{\theta}_0} f_0(\mathcal{X})^T \Theta^{-1}(I - e^{-\eta\Theta t})(f_0(\mathcal{X}) - \mathcal{Y})]\boldsymbol{\theta}_0$$
$$+ \frac{1}{d_{\boldsymbol{\theta}}}\boldsymbol{\theta}_0^T \boldsymbol{\theta}_0. \tag{9}$$

There are three terms in the equation, representing the second-order component, first-order component and the zero-order component of the changes in the Euclidean distance introduced by the gradient descent. Because in practice, $\boldsymbol{\theta}_0$ is initialized from a distribution with zero mean and small variance, the second term can be ignored. The third term is time-independent and does not affect the monotonicity in our subsequent analysis, so it can also be disregarded. We denote the second and third terms as $\epsilon$. Thus the $\ell_2$ distance can be approximated by

$$
\frac{1}{d_{\boldsymbol{\theta}}}||\boldsymbol{\theta}_t||_2^2 \approx \frac{1}{d_{\boldsymbol{\theta}}}[\nabla_{\boldsymbol{\theta}_0}f_0(\mathcal{X})^T\Theta^{-1}(I - e^{-\eta\Theta t})(f_0(\mathcal{X}) - \mathcal{Y})]^T
$$
$$
\times [\nabla_{\boldsymbol{\theta}_0}f_0(\mathcal{X})^T\Theta^{-1}(I - e^{-\eta\Theta t})(f_0(\mathcal{X}) - \mathcal{Y})] + \epsilon. \tag{10}
$$

To evaluate the evolution of the distance along the time, we evaluate its derivative of $t$ as

$$
\frac{1}{d_{\boldsymbol{\theta}}}\frac{d||\boldsymbol{\theta}_t||_2^2}{dt} = \frac{2}{d_{\boldsymbol{\theta}}}(f_0(\mathcal{X}) - \mathcal{Y})^T e^{-\eta\Theta t}(I - e^{-\eta\Theta t})(f_0(\mathcal{X}) - \mathcal{Y})\eta. \tag{11}
$$

Because all eigenvalues of $e^{-\eta\Theta t}$ are between 0 and 1, both $e^{-\eta\Theta t}$ and $I - e^{-\eta\Theta t}$ are positive definite. Because $(e^{-\eta\Theta t})(I - e^{-\eta\Theta t}) = (I - e^{-\eta\Theta t})(e^{-\eta\Theta t})$, $(I - e^{-\eta\Theta t})(e^{-\eta\Theta t})$ is also positive definite. Thus, the derivative is positive, the norm of the parameters increases along the time. $\square$

**Remark 1** (Remarks on the assumptions.)**.** *In the theorem, we use the assumptions that the width of neural network goes to infinity, which is a common assumption used in the work in neural tangent kernel (Jacot et al., 2018; Arora et al., 2019) and neural network linearization (Lee et al., 2019b). Practically, for example, the theoretical properties align well with the behaviors of architecture, like 3layer fully-connected network with layer width of 8196 (Lee et al., 2019b).*

*In the theorem, we use the assumption that the loss is MSE, which is also a common assumption used in the work in neural tangent kernel (Jacot et al., 2018; Arora et al., 2019) and neural network linearization (Lee et al., 2019b). Only with the MSE loss can the analytic solution of the dynamics of neural network training be derived.*

## D.2 Generalization to Loss with Weight Decay

**Proposition 3.** *Let $f_t$ denote a neural network trained using gradient flow on the dataset $\mathcal{D} = (\mathcal{X}, \mathcal{Y})$ at time $t$. Let $\boldsymbol{\theta}_t \in \mathcal{R}^{d_{\boldsymbol{\theta}} \times 1}$ denote the vectorized parameters of $f_t$, where $d_{\boldsymbol{\theta}}$ is the number of parameters. Let $f_0$ denote the neural network at the initialization with initial parameters $\boldsymbol{\theta}_0$. Let $\eta$ denote the learning rate. Let $\lambda$ denote the small positive balancing parameter for the weight decay term. Then, with MSE loss $\mathcal{L} \triangleq \frac{1}{2}||f(\mathcal{X}) - \mathcal{Y}||_2^2$ and weight decay $\mathcal{L}' \triangleq \frac{1}{2}||\boldsymbol{\theta}_t||_2^2$, the Euclidean norm $\boldsymbol{\theta}_t$ can be written as,*

$$
\frac{1}{d_{\boldsymbol{\theta}}}||\nabla_{\boldsymbol{\theta}_0}f_0(\mathcal{X})^T\Theta^{-1}(I - e^{-\eta\Theta t})(f_0(\mathcal{X}) - \mathcal{Y})]||_2^2 + \epsilon, \tag{12}
$$

*which increases over time $t$.*

*Proof.* Similarly based on the linearization, the dynamic of the parameters follows

$$
\dot{\boldsymbol{\theta}}_t = -\eta\nabla_{\boldsymbol{\theta}_0}f_0(\mathcal{X})^T\nabla_{f_t(\mathcal{X})}(\mathcal{L} + \mathcal{L}'). \tag{13}
$$

Under MSE loss the solution of $\boldsymbol{\theta}_t$ is

$$
\boldsymbol{\theta}_t = \boldsymbol{\theta}_0 - \nabla_{\boldsymbol{\theta}_0}f_0(\mathcal{X})^T(\Theta + \lambda I)^{-1}(I - e^{-\eta(\Theta + \lambda I)t})(f_0(\mathcal{X}) - \mathcal{Y}), \tag{14}
$$

where $I$ is the identity matrix, and $\Theta \triangleq \nabla_{\boldsymbol{\theta}_0}f_0(\mathcal{X})\nabla_{\boldsymbol{\theta}_0}f_0(\mathcal{X})^T$ is the tangent kernel at the initialization. Thus, the $\ell_2$ norm of the parameters can be written as

$$
\frac{1}{d_{\boldsymbol{\theta}}}||\boldsymbol{\theta}_t||_2^2 = \frac{1}{d_{\boldsymbol{\theta}}}[\nabla_{\boldsymbol{\theta}_0}f_0(\mathcal{X})^T(\Theta + \lambda I)^{-1}(I - e^{-\eta(\Theta + \lambda I)t})(f_0(\mathcal{X}) - \mathcal{Y})]^T
$$
$$
\times [\nabla_{\boldsymbol{\theta}_0}f_0(\mathcal{X})^T(\Theta + \lambda I)^{-1}(I - e^{-\eta(\Theta + \lambda I)t})(f_0(\mathcal{X}) - \mathcal{Y})]
$$
$$
- \frac{2}{d_{\boldsymbol{\theta}}}[\nabla_{\boldsymbol{\theta}_0}f_0(\mathcal{X})^T(\Theta + \lambda I)^{-1}(I - e^{-\eta(\Theta + \lambda I)t})(f_0(\mathcal{X}) - \mathcal{Y})]\boldsymbol{\theta}_0
$$
$$
+ \frac{1}{d_{\boldsymbol{\theta}}}\boldsymbol{\theta}_0^T\boldsymbol{\theta}_0. \tag{15}
$$

Collecting the high order term into $\epsilon$, the $\ell_2$ norm can be approximated by

$$
\frac{1}{d_{\boldsymbol{\theta}}}||\boldsymbol{\theta}_t||_2^2 \approx \frac{1}{d_{\boldsymbol{\theta}}}[\nabla_{\boldsymbol{\theta}_0}f_0(\mathcal{X})^T(\Theta + \lambda I)^{-1}(I - e^{-\eta(\Theta+\lambda I)t})(f_0(\mathcal{X}) - \mathcal{Y})]^T
$$
$$
\times [\nabla_{\boldsymbol{\theta}_0}f_0(\mathcal{X})^T(\Theta + \lambda I)^{-1}(I - e^{-\eta(\Theta+\lambda I)t})(f_0(\mathcal{X}) - \mathcal{Y})] + \epsilon. \tag{16}
$$

Its derivative of $t$ can be written as

$$
\frac{1}{d_{\boldsymbol{\theta}}}\frac{d||\boldsymbol{\theta}_t||_2^2}{dt} = \frac{2\eta}{d_{\boldsymbol{\theta}}}(f_0(\mathcal{X}) - \mathcal{Y})^T\nabla_{\boldsymbol{\theta}_0}f_0(\mathcal{X})^T(\Theta + \lambda I)^{-1}e^{-\eta(\Theta+\lambda I)t}
$$
$$
\times [I - [e^{-\eta(\Theta+\lambda I)t}]^{-1}](\Theta + \lambda I)e^{-\eta(\Theta+\lambda I)t}(\Theta + \lambda I)^{-1}\nabla_{\boldsymbol{\theta}_0}f_0(\mathcal{X})(f_0(\mathcal{X}) - \mathcal{Y}).
$$
$$\tag{17}$$

Because $\lambda$ is small the term $I - [e^{-\eta(\Theta+\lambda I)t}]^{-1}$ can be approximated as $(\Theta + \lambda I)t$. Similarly to the discussion in Appendix D.1, the derivative is positive definite. Thus, the derivative is positive, the norm of the parameters increases along the time. $\square$

## E  FURTHER DISCUSSIONS

### E.1  IMPACTS, LIMITATIONS, AND FUTURE EXPLORATIONS

Neural phylogeny detection is the first task that identifies fine-tuning pairs and directions directly from a collection of models without relying on generational annotations. Identifying the phylogeny of models has significant implications for research and model governance. The information about model relationships can be used to study the inheritance of knowledge, and the spread of bias and weaknesses among models. Practitioners can use detected phylogeny to regulate model usage. Since phylogeny detection only requires parts of the model parameters, extracting the knowledge learned by the models or related information used for training is difficult, thus minimizing the risk of privacy or intellectual property breaches. However, the ability of phylogeny detection to uncover fine-tuning relationships between models can be applied to research on model attacks, defenses, and privacy extraction and protection. For example, when designing black-box attack methods, a more effective proxy model can be selected from a model library using phylogeny detection.

As the first attempt to address a novel task, our solutions have many areas for further exploration. First, due to the reliance on the natural alignment between parent and child model parameters, our models cannot handle fine-tuning with additional parameters introduced or changes in neural network structures. However, this strategy is widely used in the efficient tuning of large models, making it highly relevant for practical applications and a promising direction for future research. On the other hand, our learning-free method treats phylogeny detection as a clustering task, which is highly dependent on data structure and sensitive to hyperparameter design. Although our learning-free method achieved good results, in practical applications, it is worth exploring how to introduce more effective hyperparameter selection mechanisms or approach the phylogeny detection problem from different perspectives to avoid relying on clustering methods.

### E.2  PHYLOGENY DETECTION AND ATTACKS

Our method requires access to model parameters and assumes that both the parent and child models are fully accessible. Therefore, our finetuning relationship detection is primarily designed for open-source models, such as the publicly available models on HuggingFace. For these models, white-box attacks can be performed directly with high success rate.

However, the proposed Phylogeny Detection does not enhance the white-box attack. While it may enhance black-box and gray-box attacks by improving the selection of proxy models, these improved attacks will still be less efficient and effective than white-box attacks.

On the other hand, the issue of improved proxy model selection leading to more effective black-box and gray-box attacks will only arise if black-box or gray-box finetuning relationship detection becomes feasible. We recognize black-box or gray-box finetuning relationship detection as an important direction for future research and will address it in the Future Work and Limitations section.

