# OpenReview forum: "Neural Phylogeny: Fine-Tuning Relationship Detection among Neural Networks"
_ICLR.cc/2025/Conference — ICLR 2025 Poster_

### Official Review · Reviewer_FjhM · 2024-10-30

**Soundness:** 2
**Presentation:** 3
**Contribution:** 2
**Rating:** 6
**Confidence:** 4

**Summary:**

The paper presents an algorithm to identify a relationship between networks in the form parent-children pair. In contrast to the existing neural lineage detection, the proposed algorithm proposed a new notion termed neural phylogeny where the child and parent model can have the same structure. The paper proposes a learning-free neural phylogeny detection method. The experiments are performed using multiple shallow and deep models.

**Strengths:**

+ The paper presents an effective approach for the identification of parent-children relationships among the deep learning models.

+ The paper is easy to read and follow.

**Weaknesses:**

- The primary weakness of the proposed research is the lack of motivation and application of the current work. What is the use of neural phylogeny and where it can be applied?

- Why the fake initialization has been used to compute the distance? Is the proposed metric sensitive to this initialization?

- The paper mentions "fine-tuned child model undergoes a longer gradient optimization". Is the phylogeny depend on the number of fine-tuning steps/processes?

- How does the fake random initialization help in finding the fine-tuning direction?

- Comparison with neural lineage algorithms (discussed in section 2.1) must be performed. Along with the graph link prediction algorithm also needs to be added.

- Another primary limitation is the limited success of the proposed algorithm (especially ResNet and ViT) on the low-resolution color images (CIFAR10- 5/50 shots). It raises the question of the success of different deep transformers and CNNs on high-resolution datasets such as CIFAR100, ImageNet, and Tiny ImageNet on different shot settings.

**Questions:**

- The primary weakness of the proposed research is the lack of motivation and application of the current work. What is the use of neural phylogeny and where it can be applied?

- Why the fake initialization has been used to compute the distance? Is the proposed metric sensitive to this initialization?

- The paper mentions "fine-tuned child model undergoes a longer gradient optimization". Is the phylogeny depend on the number of fine-tuning steps/processes?

- How does the fake random initialization help in finding the fine-tuning direction?

- Comparison with neural lineage algorithms (discussed in section 2.1) must be performed. Along with the graph link prediction algorithm also needs to be added.

- Another primary limitation is the limited success of the proposed algorithm (especially ResNet and ViT) on the low-resolution color images (CIFAR10- 5/50 shots). It raises the question of the success of different deep transformers and CNNs on high-resolution datasets such as CIFAR100, ImageNet, and Tiny ImageNet on different shot settings.

---

### Official Review · Reviewer_MurH · 2024-10-31

**Soundness:** 3
**Presentation:** 3
**Contribution:** 3
**Rating:** 6
**Confidence:** 4

**Summary:**

This paper proposes an interesting concept Neural Phylogeny, which aims to determine the parent-child relationship in a given set of models, which can play a role in model copyright protection in context of fine-tuning. Simply speaking, this article determines the length of the training cycle based on the distance from the model weight to the initial weight, which can be used as evidence to determine the parent-child relationship. Based on this idea, this paper proposes two specific methods and conducts comprehensive tests.

**Strengths:**

1. The concept of neural phylogeny detection is a new, well-defined task, advancing beyond neural lineage to focus on detecting both the existence and direction of fine-tuning relationships among neural networks. This novel concept is valuable for understanding model evolution and intellectual property tracking​.

2. The design motivation of the solution is well-founded with theoretical analysis and empirical evidence.

3. Extensive evaluations are conducted to indicate the superiority of the proposed methods in detecting fine-tuning relationships.

**Weaknesses:**

1. The proposed method is only applicable to relationship traceability in white-box scenarios. However, in practical applications, especially in IP-sensitive environments, only partial model information may be available, and the effectiveness of the proposed methods under such constraints was not evaluated​.

2. Whether different sampling methods of the initial weight will affect the detection results needs to be discussed.

3. I note that the performance of the proposed method varies greatly across architectures and datasets (see Table 2). It is worth exploring the reasons behind this unstable detection and whether this instability is common or unique to the proposed method.

4. The learning-based approach achieved high accuracy on tested models but might overfit specific architectures or datasets used for training. This raises concerns about generalizability, particularly when applying the model to untested or proprietary architectures not represented in the current experiments​.

**Questions:**

None

**Details Of Ethics Concerns:**

N.A.

---

### Official Review · Reviewer_WN3r · 2024-11-02

**Soundness:** 3
**Presentation:** 3
**Contribution:** 2
**Rating:** 6
**Confidence:** 4

**Summary:**

This paper proposes a new problem, neural phylogeny, which aims to predict the fine-tuning relationship of models. Specifically, given a set of neural networks, the authors predict clusters of models such that all models were fine-tuned from a common pre-trained (or parent) model. Furthermore, they predict which model is the parent (pre-trained model) and which are the children (fine-tuned models). The authors first define a method for predicting the parent model in a given (parent, child) pair. Next, they propose a few methods for predicting model clusters and parents within those clusters, including unsupervised clustering methods and a supervised neural network method. They perform experiments on several model architectures and fine-tuned datasets and find that the supervised neural network method is the top performing method.

**Strengths:**

1. The problem of neural phylogeny is an interesting concept and may be more practical than the previously defined neural lineage detection because (a) it is data-free (i.e., does not require access to the fine-tuning dataset) and (b) it does not require prior knowledge of whether a model is a model is a parent or child.
2. The detection model used for Figure 1 and the metric for determining the fine-tuning relationship between two models (proposition I) is novel and the empirical results show these are effective.
3. The paper considers multiple methods for predicting clusters and parent node, including unsupervised and supervised methods. Additionally, the authors test their methods on several model architectures with varying sizes and types of layers (FCNet, ConvNet, ResNet, ViT, Llama, Diffusion models).
4. The paper is well-written.

**Weaknesses:**

1. These methods require that the practitioner has access to all the network weights. Furthermore, the supervised model "Phylogeny Detector" requires having access to a training set that includes the weights of the parent model and several children models. These assumptions do not seem practical and the unsupervised performance is poor.
2. The methods that the authors propose for solving the neural phylogeny problem require that all of the node models have the same model architecture (i.e., if the fine-tuned model has additional layers or parameters, these methods won't work).
3. The experimental design and results are hard to understand, even after reading the relevant sections in the appendix.

**Questions:**

Questions
1. Can you please clarify how you found the results in Table 2? Specifically, are models trained on the same dataset with different learning rates and/or batch sizes considered to be different parent models? Are the results averaged over the 8 seeds mentioned in the appendix or something else?
1. In equation 2, why do you select the model that is most frequently identified as the parent instead of just taking the model with the minimum distance to $\mathbf{z}$ (i.e., $\arg\min_{i} dist(\boldsymbol{\theta}^{(i)}, \mathbf{z})$)?
2. Why did you choose to use a transformer model over other supervised models?
3. I assume "Uniform" and "Gaussian" in Figure 2 refer to the random distributions used for $\mathbf{z}$. On line 256-258, you say that "this theorem only requires z to have zero mean, which makes the choice of proxy distribution robust when calculating the distance, without the need to precisely know the initial parameter values of the neural network". Have you validated this empirically with other initialization techniques that are common for neural networks, like Kaiming or Xavier?

Suggestions
1. To understand Figure 1, we must read the Appendix. I think this is a main contribution of the paper and the experimental details should be clearly stated in the main body. Additionally, the metrics (Overall acc, lineage acc, and direction acc) should be more clearly defined, preferably using math notation.
2. In Figures 2 and 3 the legends are unclear. I suggest replacing legend titles like "color" and "line style" with more useful explanations and clarifying method names ("Along clustering" and "After Clustering") as you define them.
4. The notation in Figure 4 does not match the notation in the main text.

---

### Official Review · Reviewer_fSGg · 2024-11-03

**Soundness:** 2
**Presentation:** 3
**Contribution:** 2
**Rating:** 6
**Confidence:** 3

**Summary:**

This study introduces the concept of neural phylogeny detection, which is a new task, focusing on identifying the parent-child relationship between neural networks and determining the direction of fine-tuning between them. These two methods have been tested in different model architectures and learning environments, and proved to be effective and reliable in detecting the fine-tuning relationship of transgenerational neural networks.

**Strengths:**

- Novelty of neural relationship detection: this work provides a groundbreaking method for understanding the inheritance of knowledge and connections in the growing neural model network, and solves the key challenge of deep learning traceability.

- Dual methods of flexibility and accuracy: by providing learning free and learning based methods, this research provides the option of giving consideration to easy implementation and high detection accuracy to meet various use cases.

- Robust evaluation: a large number of experiments in various network architectures and learning scenarios have confirmed the universality and robustness of the proposed method, showing the applicability of the framework to different model structures and generational relationships.

**Weaknesses:**

**Some major comments:**

- The metric relies on a “fake initialization” since the true initialization parameters are unavailable. This introduces a level of approximation that may impact the accuracy, particularly in scenarios where the proxy distribution used for fake initialization does not align closely with the true initialization distribution.

- Integrating fine-tuning direction into clustering algorithms introduces added complexity, especially with algorithms like KMeans and MeanShift. Adjusting the clustering to identify the parent-child direction requires modifications that could add computational cost and reduce interpretability of traditional clustering processes.

- The effectiveness of the clustering approach depends heavily on the probability parameter $\alpha$. The need for empirical tuning of $\alpha$ to achieve optimal results adds another layer of complexity, which may limit the method’s generalizability across different datasets and model types.

- The proposed methods require iterating across high-dimensional parameter spaces and maintaining computational efficiency, which may be challenging with larger networks or model sets, potentially impacting scalability and real-world applicability.

**Some minor comments:**

- Related work should introduce the differences between this paper and previous work and explain the advantages of this research over previous work. It should not only list the work.

- The objective of this study has not been clearly described (in Section 3.1)

**Questions:**

- How robust is the proposed fake initialization metric across different types of neural architectures beyond ResNet, Stable Diffusion, and LLaMA? Has it been tested with a variety of non-fully connected networks or transformers?

- Given that the choice of $\alpha$ impacts clustering accuracy, could a systematic method be developed for selecting an optimal $\alpha$ value without relying on empirical tuning?

- How does the proposed framework handle situations where multiple child models are fine-tuned from a single parent model, or when models have multiple parent models (e.g., ensembling scenarios)?

- Would the approach be applicable or require modifications when dealing with pruned or compressed models, which might have distinct parameter distributions from standard fine-tuned models?

- The author used fmnist, emnist, cifar10, svhn as datasets in the experimental part. However, the structure of these datasets is too simple. Can the author prove the method and rationality proposed in this paper?

---

> ### Comment · Reviewer_fSGg · 2024-11-25
> **Reply to Submission1414 Authors**
>
> Thank you for the prompt response. I think the authors' response has answered all my doubts. Based on this, I have decided to modify my rating to 5.
>
> Thanks.

---

### Meta-Review · Area_Chair_qrLJ · 2024-12-20

**Metareview:**

This paper introduces a novel task, neural phylogeny detection, aimed at identifying the existence and direction of fine-tuning relationships between neural networks. The task is both interesting and practical, with potential applications in areas like copyright protection. The paper proposes two approaches for this task: a learning-free method and a learning-based method. Extensive experimental results demonstrate the effectiveness of both methods. All four reviewers voted for borderline acceptance. Given the novelty and practicality of the task, I recommend acceptance.

**Additional Comments On Reviewer Discussion:**

The main concerns raised by the reviewers relate to the details and effects of fake initialization, the limitations of accessing all model weights, and similar issues. During the rebuttal stage, the authors provided detailed responses, including supporting empirical results. All reviewers expressed satisfaction with the authors’ clarifications.

---

### Decision · Program_Chairs · 2025-01-22

Accept (Poster)